# Multi-Omics Model Applied to Cancer Genetics

**DOI:** 10.3390/ijms22115751

**Published:** 2021-05-27

**Authors:** Francesco Pettini, Anna Visibelli, Vittoria Cicaloni, Daniele Iovinelli, Ottavia Spiga

**Affiliations:** 1Department of Medical Biotechnology, University of Siena, Via M. Bracci 2, 53100 Siena, Italy; 2Department of Biotechnology, Chemistry and Pharmacy, University of Siena, Via A. Moro 2, 53100 Siena, Italy; anna.visibelli@student.unisi.it (A.V.); daniele.iovinelli@student.unisi.it (D.I.); ottavia.spiga@unisi.it (O.S.); 3Toscana Life Sciences Foundation, Via Fiorentina 1, 53100 Siena, Italy; v.cicaloni@toscanalifesciences.org

**Keywords:** data analysis, artificial intelligence, precision medicine, machine learning models, computational oncology, cancer disease, omics tools

## Abstract

In this review, we focus on bioinformatic oncology as an integrative discipline that incorporates knowledge from the mathematical, physical, and computational fields to further the biomedical understanding of cancer. Before providing a deeper insight into the bioinformatics approach and utilities involved in oncology, we must understand what is a system biology framework and the genetic connection, because of the high heterogenicity of the backgrounds of people approaching precision medicine. In fact, it is essential to providing general theoretical information on genomics, epigenomics, and transcriptomics to understand the phases of multi-omics approach. We consider how to create a multi-omics model. In the last section, we describe the new frontiers and future perspectives of this field.

## 1. Introduction

Last fact sheets from World Health Organization (WHO), updated to March 2021, reports cancer is the second leading cause of death worldwide, accounting for nearly 10 million deaths in 2020. Approximately 70% of the deaths from cancer occur in low- and middle-income countries. Breast, lung, colorectal, and prostate cancers are the most common [1].

A correct cancer diagnosis is essential for adequate and effective treatment because every tumor is involved in interactions with non-cancer elements such as gene-environment interactions (GxE), micro-environmental interactions, and those with the immune system; intercellular interactions within the tumor environment; and intracellular interactions, such as transcriptional regulation and gene co-expression, signaling and metabolic pathways, as well as protein interactions (Figure 1) [2].

This is the reason why only an integrating framework among different omics layers can gather and organize the knowledge gained with each experimental approach into mechanistic or semi-mechanistic descriptions of the biological phenomenon [3].

Multi-omics model is defined as a biological approach that, by using one or more current high-throughput experimental techniques, can investigate physiological or pathological phenomena and characterize biomolecular systems at different levels. As a matter of fact, each omics contributes on a specific fashion to shape the actual biological phenotype of interest. 

Thus, a comprehensive recognition of molecular networks based on multi-omics data has an important scientific role to understand the molecular mechanisms of cancer, but this is possible only because of bioinformatics application [4]. Computational oncology can be defined as an integrative discipline incorporating scientific backgrounds from the mathematical, physical, and computational fields to get a deeper understanding on malignancies [2].

In the coming age of omics technologies, next gen sequencing, proteomics, metabolomics, and other high throughput techniques will become the usual tools in biomedical cancer research. However, their integrative approach is not trivial due to the broad diversity of data types, dynamic ranges and sources of experimental and analytical errors characteristic of each omics [2]. The multi-omics systematic study of cancer found many different factors involved in the development/maintenance of the malignant state such as genetic aberrations, epigenetic alterations, changes in the response to signaling pathways, metabolic alterations, and many others [5]. The advent of high-throughput technologies has permitted the development of systems biology. The system biology paradigm tries to analyze cancer as a complex and intricate pathology and to gain insight into its molecular origin by taking into account the different contributions like DNA mutations, deregulation of the gene expression, metabolic abnormalities, and aberrant pathway signaling [2].

The essential basis of systems biology is to consider a biological phenomenon as a system of interconnected elements such as many complex molecular and environmental components interacting with each other at different levels. For example, tumor behavior is determined by a combination of changes in genomic information possibly associated with abnormal gene expression, protein profiles, and different cellular pathways. In this scenario, the complex interaction of DNA and proteins in replication, transcription, metabolic, and signaling networks are considered the decisive causes for cancer cells dis-functioning [2]. The integration of multi-omics data provides a platform to connect the genomic or epigenomic alterations to transcriptome, proteome, and metabolome networks underling the cellular response to a perturbation. Powerful and sophisticated computational tools can identify the interconnection between genomic aberrations with differentially expressed mRNAs, proteins, and metabolites associated with cancer-driven cellular perturbation [6]. If on the one hand this aspect provides an opportunity to better study the cellular response, on the other hand it poses a challenge for systems biology-driven modelling. Therefore, the next step of systems biology approach focuses on dynamic models that can deal with thousands of mRNA, protein, and metabolite changes developing effective strategies to administer personalized cancer therapy [7]. Summarizing, the main goal of the systems biology research driven by multi-omics data is to develop predictive models that are refined by experimental validations in order to select patients based on personalized multi-omics data and stratifying them to determine who are most likely to benefit from targeted therapies [6,8].

Definition and detection of cancer-distinctive features allow the investigation of the transition process of a normal cell to malignancy. Generally, the hallmarks involve phenotypic and molecular changes in several metabolic pathways such as uncontrolled proliferation by blocking growth suppressors, reprogramming of energy metabolism, evading immune destruction, resisting cell death, angiogenesis, and metastasis [9]. These variations in cellular machinery are driven by molecular aberration in several omics layers such as genome, epigenome, transcriptome, proteome, and metabolome within cancer cells. Specifically, by applying next generation sequencing to cancer cell genomes, it is possible to reveal how mutations in proliferative genes like B-raf drives the activation of mitogen-activated protein- (MAP-) kinase signaling pathway underlying an uncontrolled cell proliferation [10]. Molecular aberrations leading to cancer are involved not only in genomic mutational events but also in the epigenome. In particular, aberrant epigenetic mechanisms can be responsible for silencing of certain cancer suppressor genes [11]. The multistep processes of invasion and metastasis require a transition of epithelial cell toward mesenchymal phenotype to colonize distant sites. Recent studies have revealed that epithelial-mesenchymal transition is induced by specific transcription factors that coordinate the invasion and metastasis processes [9]. By applying transcriptomics techniques it is possible to investigate the transcription factors involved in transcription regulatory networks assumed to be activated in malignancy. Moreover, manifestations of cancer hallmarks also affected cellular metabolism, in fact tumor cells can reprogram glucose metabolism and energy production pathways detectable with a metabolomics approach [6].

## 2. Genomics and Molecular Processes

### 2.1. Cancer Gene Types

In general, cancer disrupts cellular relations and results in the dysfunction of vital genes. This disturbance is affective in the cell cycle and enhances abnormal proliferation [12,13]. There are three main types of cancer genes that control cell growth and can cause cancer to develop:Oncogenes. These, when mutated, actively promote cell proliferation. They are formed when proto-oncogenes that promote cell division are improperly activated, so they are not known to be inherited. They may lead to increased/dysregulated expression of the gene in a new location or to production of fusion proteins with new functions [14]. Two common oncogenes are HER2 and RAS.Gatekeeper genes. These are protective genes, also known as tumor suppressor genes. Normally, they negatively control cell growth by monitoring and controlling the cell phases or repairing mismatched DNA. Autosomal recessive mutations in tumor suppressor gene cause loss of function effect at the cellular level, inducing cells to grow uncontrollably, which may eventually form a tumor. Examples of tumor-suppressor genes include BRCA1, BRCA2, and p53 or TP53. Germline mutations in BRCA1 or BRCA2 genes increase a woman’s risk of developing hereditary breast or ovarian cancers and a man’s risk of developing hereditary prostate or breast cancers. They also increase the risk of pancreatic cancer and melanoma in women and men [15]. The most mutated gene in people with cancer is p53 or TP53. More than 50% of cancers involve a missing or damaged p53 gene. Most p53 gene mutations are acquired. Germline p53 mutations are rare, but patients who carry them are at a higher risk of developing many different types of cancer [15].Carekeeper genes. These fix the mistakes made when DNA is copied. Many of them function as tumor suppressor genes. BRCA1, BRCA2, and p53 are all DNA repair genes. If a person has an error in a DNA repair gene, mistakes remain uncorrected. Then, the mistakes become mutations. These mutations may eventually lead to cancer, particularly mutations in tumor suppressor genes or oncogenes. Mutations in DNA repair genes may be inherited or acquired. Lynch syndrome is an example of the inherited kind. BRCA1, BRCA2, and p53 mutations and their associated syndromes are also inherited [14].

As said before genetic changes that promote cancer can be inherited from our parents if the changes are present in germ cells, which are the reproductive cells of the body (eggs and sperm). Such changes, called germline changes, are found in every cell of the offspring. Cancer-causing genetic changes can also be acquired during one’s lifetime and are called somatic (or acquired) changes [14]. Next, we will take these aspects into consideration.

### 2.2. Genomic Instability

Somatic mutations, based on their function, involves driver mutations, conferring growth advantage to the cancer cells. Otherwise, acquired mutations do not confer any growth advantage to the cancer cells nor contribute to cancer development [16]. Chromosomal changes are highly variable, they can be grouped into two general categories [17]:Balanced structural changes; the genetic material is equally exchanged, even if genetic information was rearranged into an abnormal gene;Unbalanced or nonreciprocal structural changes; the exchange is not equally distributed, and genetic material is added or lost. This can range from the loss or gain of a single base pair to the loss or gain of the entire chromosomes.

From Knudson’s two hits hypothesis [18] studies to the present, scientists suggested that the primary pathogenetic changes in cancer result from balanced rearrangements, while the secondary hits that occur during cancer progression are from unbalanced changes (see Table A1).

In Wilms’ tumor and retinoblastoma, gene deletions or inactivation are responsible for cancer development [17]. Deletions, inversions, and translocations are commonly detected in the Philadelphia chromosome, the first balanced chromosomal mutation described in cancer cells; it is the result of a reciprocal translocation between chromosomes 9 and 22 with breakpoints in the c-abl gene on chromosome 9 and the c-bcr gene on chromosome 22. The fusion gene created by this rearrangement encodes a tyrosine kinase that promotes cancer in white blood cells (chronic myeloid leukemia) [19]. Burkitt’s lymphoma is another type of cancer associated with reciprocal translocations involving chromosome 8 and a chromosome carrying an immunoglobulin gene (2, 14, or 22). The translocations juxtapose c-myc to the genes for the immunoglobulin genes, causing overexpression of c-myc in B cells. The c-myc gene encodes a transcription factor that activates genes for cell division [20]. Large portions of chromosomes can also be lost, as occurred on chromosomes 1p and 16q in solid tumor cells [17]. Gene duplications and increases in gene copy numbers can also contribute to cancer and can be detected, for example, in many sarcomas. The chromosomal region 12q13-q14 encodes a binding protein called MDM2, which is known to bind to a tumor suppressor called p53. Amplification of MDM2 prevents p53 from regulating cell growth, which can result in tumor formation. Also, mutations in carekeeper genes can additionally lead to rearrangements and duplications [17]. Most of the cancers harbor more than one driver gene mutation. Breast, colorectal, and prostate cancers require from five to seven driver mutations for cancer initiation and progression, while hematological malignancies may require fewer. TP53, RB1, EGFR, and KRAS, are widely known mutated genes in various cancer types, whereas others are rare and/or restricted to one cancer [16].

### 2.3. Epigenomic Instability

It is largely proved that genomic instability is a reductive model; studies demonstrated epigenetic errors resulting in aberrant gene silencing/activation [21]. According to the definition, epigenetics is a dynamic situation in the study of cell fate, that alter the structure of DNA without directly affecting and mutating its sequence [22]. In fact, mutations occurred in the elements that regulate the expression or repression of the genome, such as transcription factors and noncoding RNAs, with a consistent effect on the coordination of multiple biological processes. These elements can be divided into three roles: “writers” and “erasers” refer to enzymes that transfer or remove chemical groups to or from DNA or histones, respectively; “readers” are proteins that can recognize the modified DNA or histones [23].

In tumor tissues, different tumor cells show various patterns of histone modification, genome-wide or in individual genes, demonstrating that epigenetic heterogeneity exists at a cellular level and suggesting that tumorigenesis is the consequence of the combined action of multiple epigenetic events [24]. For example, the repression of gatekeeper genes is usually caused by DNA modification in the methylation of CpG islands together with hypoacetylated and hypermethylated histones [25]. Gene silencing experiments identified several hallmarks of epigenetic events, including histone H3 and H4 hypoacetylation, histone H3K9 methylation, and cytosine methylation [26]. Major epigenetic modifications are classified as DNA modifications, histone modifications, effects of non-coding RNA (Figure 2).

DNA methylation typically occurs at CpG sites (cytosine-phosphate-guanine) sites. This methylation results in the conversion of the cytosine to 5-methylcytosine. The formation of Me-CpG is catalyzed by enzymes called DNA methyltransferases (DNMTs). This modification is common in body cells; in tumors, we can observe an hypomethylation of the genome [27] that results in initiate and propagate oncogenesis, by inducing chromosome instabilities and transcriptional activation of oncogenes and pro-metastatic genes, such as *r-ras* [28]. This state is accompanied by a region- and gene-specific hypermethylation of multiple CpG islands [29,30]. Hypermethylation of CpG islands in the promoter region of a tumor suppressor or otherwise cancer-related gene is often associated with transcriptional silencing of the related gene. Numerous genes associated to various pathways are known and rapidly identified; actually, genes involved in signal transduction (*APC*), DNA repair (*MGMT*, *MLH1*, *BRCA1*), detoxification (*GSTP1*), cell cycle regulation (*p15*, *p16*, *RB*), differentiation (*MYOD1*), angiogenesis (*THBS1*, *VHL*), and apoptosis (*Caspases*, *p14*, *DAPK*) are reported and largely studied [31] (for a complete overview on key regulatory factors of DNA methylation in cancer we suggest to see Table 1 of [23]). DNA methylation can act as one hit having the same functional effect as a genetic point mutation, as proven by numerous experiments in which re-establishing expression of tumor suppressor genes could be reached through drugs inducing demethylation. Epimutations can inactivate one of the two alleles, while the other is lost through genetic mechanisms or silence both alleles [32]. For example, a study conducted on 50 RB patients (45 unilateral and 5 bilateral), selected from an initial cohort of 476 RB cases diagnosed over a period of 17 years at the Retinoblastoma Referral Centre of Siena (Ophthalmology Department, AOUS), provided evidence supporting the identification of a constitutional epimutation acting as the first “hit” in the Knudson model of RB development and suggests that epimutations do not represent a frequent cause of RB predisposition but this is an understudied etiological phenomenon and, besides promoter methylation, other untested epigenetic events may reduce gene expression, phenocopying RB onset [33]. Epigenetic changes occur at higher frequency with respect to genetic changes and might be especially important in the first phase of human neoplasia; aberrant promoter methylation is initiated at ~1% of all CpG islands and as much as 10% become methylated during the multistep process of tumorigenesis [34]. As a stable nucleic-acid-based modification with limited dynamic range that is technically easy to handle, DNA methylation is a promising biomarker for non-invasive detection of different tumor types [35,36,37,38]. Besides early detection, the methylation status of CpG islands can be used to characterize and classify cancers. While for example, breast, or testicular tumors show global low levels of methylation, some other tumor types such as colon tumors, acute myeloid leukemias, or gliomas are characterized by high levels of methylation, although some heterogeneity is observed in almost all tumor types. So, methylation patterns can be an important hallmark to identify and classify the different types of human cancers [34,39]. DNA methylation profile can be used also to predict and monitor the response to anti-neoplastic treatment [39,40]. 

Histones are made up of amino acids, like all other proteins. Amino acids located in the tail of them are targets for enzymes that attach or remove chemical markers, therefore the potential main site of histone modifications, in particular, lysine (Lys) and serine (Ser) are common targets. Histone modification is a relatively complicated process compared to DNA methylation that involves only two types of enzymes, which can add or remove only methyl groups to cytosine. When histone modification patterns are altered, it can lead to unregulated activity or silencing of genes related to cancer onset [24,41]. 

All families of protein involved in chromatin remodeling pathways are associated with cancer, although in most cases, the molecular mechanisms underlying their functions remain unknown [42]. Overall reduction of mono acetylated H4K16 forms the majority of histone modifications in cancer cells [43]. Other modifications, as histone H3 acetylation and methylations, interfere with the chromatin remodeling status, leading to repression or activation of transcription [44]. In contrast ubiquitination is a larger covalent modification, commonly related to the H2B. H2BK123ub1 modification involves the addition of ubiquitin chain to histone H2B and this modification results in regulating transcriptional initiation and elongation, while H2AK119ub1 is involved in gene silencing [45]. Similarly, phosphorylated forms of histones, H3S10ph and H2BS32ph, are implicated in the expression of proto-oncogenes, such as MYC, JUN, and FOS [46] (for a more detailed overview on important enzymes or proteins that regulate histone modification we suggest to see Tables 2 and 3 in reference [23]). 

Epigenetic-related noncoding RNAs (ncRNAs) include microRNAs (miRNAs), small interfering RNA (siRNAs), Piwi-interacting RNA (piRNAs), and long noncoding RNAs (lncRNAs). MiRNAs, one of the most studied ncRNAs, are small RNAs (from 19 to 22 nucleotides in length) known to influence gene expression by way of targeting messenger RNA (mRNA) [24]. Generally, they can be classified into tumor-promoting and tumor-suppressing miRNAs. In fact, during tumorigenesis we can observe that oncogenic miRNAs such as miR-155, miR-21, and miR-17-92 are usually overexpressed, while miRNAs such as miR-15-16 are downregulated. There is another type of miRNA, cellular context-dependent miRNAs, working in tumorigenesis. For example, miR-146 has been shown to be overexpressed in multiple cancers, whereas a study of Garcia et al. has proven that miR-146 can reduce the expression of BRCA1. At the same time, the expression of proteins and enzymes is also regulated by certain miRNAs. The miR-101 reduces EZH2 expression, and abnormal downregulation of miR-101 has been observed in several types of cancers [24]. As miRNAs play critical roles in regulating functions of the cells, disruption in their structure and turnover can also cause diseases [47]. CLL is the first human disease that is associated with miRNA disorders [48]. miRNAs can be used as cancer diagnosis biomarker as determinant of cancer prognosis and patient overall survival. MiRNAs can be used to classify myeloid malignancies. For example, in a study of meta-analysis performed by Erdogan et al. [49], they identified 13 miRNAs of interest from a total of 42 MDS samples and 45 controls studied, 8 of which proved statistically significant on real-time polymerase chain reaction verification. LncRNAs are another diffused group of ncRNAs that can play an important role in tumorigenesis. Some of them are cancer type-specific, such as PCGEM1 in prostate cancer and HEIH in hepatocellular carcinoma. Many aberrant lncRNAs have been discovered in various cancers; for example, dysregulation of HOTAIR has been found in lung, pancreatic, and colorectal cancer [24]. ncRNAs can either be directly involved in tumorigenesis or indirectly affect tumor development by participating in other epigenetic events [24].

## 3. Roles of Computational Approach in Multi-Omics Era

Computational approach plays central roles not only in the analysis of high-throughput experiments, but also in data acquisition, in processing of raw file derived from several instruments, in storage and management of large streams of omics information and in the data model integration. Bioinformatics workflow management systems can be used in developing and in application of a certain pipeline. Examples of such systems include Galaxy [50], Snakemake [51], Nextflow [52], and the general-purpose Common Workflow Language [53]. Several tools for omics data studies are available in Bioconductor project as packages for the R language [54] and in Biopython project [55].

### 3.1. Data Acquisition

All the omics technologies have a specific role to figure out the complex phenotype of cells especially in complex diseases like cancer. Knowledge of the biological molecular basis of different cellular signaling pathways does not involve only genes and transcripts, in fact, proteins and metabolites are particularly important to predict the phenotypic alterations for diagnosis and prognosis of cancer, and for this reason, in this chapter, we will spend some words about them. Table 1 represents a summary of the applications of different NGS-based and mass spectrometry-based techniques which are at the basis of different omics data acquisition approaches.

#### 3.1.1. Genomics

To date, genomics approach has highly sustained the finding and investigation of variations at both the germline and somatic levels thanks to many progresses in genome-exome sequencing techniques, for instance from the Sanger sequencing-based approaches to the NGS-based sequencing. Bioinformatics has always had a central role in the analysis of downstream genetic data. For example, in the multiscale scale project “The Cancer Genome Atlas” (TCGA), researchers used NGS sequencing associated to bioinformatics tools with the aim to discover somatic mutational landscape across thousands of tumor samples and to understand the complexity underlying different cancer types [56,57]. For the analysis of NGS data a sequence aligner tool is used on the sequence data (stored in FASTQ format). Some popular aligners are the stand-alone BWA [58], Bowtie [59], Bowtie2 [60], and SNAP [61], with aligned sequences being stored in SAM (Sequence Alignment Map, text-based) or BAM (Binary Alignment Map) files.

#### 3.1.2. Epigenomics

Epigenomics is concerned with the genome-wide identification of chemical modifications (i.e., methylation and acetylation of DNA) which are involved in regulatory mechanisms controlling gene expression and cellular phenotypes [62]. Chromatin immunoprecipitation (ChIP) assays-coupled NGS (ChIP-seq) and methylation analysis through whole-genome bisulfite sequencing (WGBS) or bisulfite sequencing (BSSeq) are the most widely used methods in epigenomics analysis [6]. By exploiting the advances in NGS field, it is now possible to analyze genome-wide methylome patterns at a single nucleotide resolution and to detect the methylated cytosine bases in genomic DNA. Data from array-based techniques can be analyzed using dedicated packages such as *methylationArrayAnalysis* [63], whereas for ChIP-seq data processing tools like SICER2 [64], PeakRanger [65], GEM [66], MUSIC [67], PePr [68], DFilter [69], and MACS [70] are used.

#### 3.1.3. Transcriptomics

The detection and quantification of RNA transcripts (mRNA, noncoding RNA and microRNAs) is possible owing to the employment of several transcriptomics techniques. Differently from the static nature of genome, transcriptome dynamically changes as consequence of temporal cellular and extracellular stimuli. Microarray was the technique of choice to detect alterations in cellular mRNA levels in a high-throughput manner owing to its ability to quantify the relative abundance of mRNAs for thousands of genes at the same time. Microarrays are widely used to facilitate the identification of genes with differential expression between normal and cancer conditions. With the advent of NGS, the identification of the presence and the abundance of RNA transcripts in genome-wide manner became possible. In contrast to microarrays technique, RNA-seq does not depend on the transcript-specific probes and thus can effectively perform an unbiased detection of novel transcripts, also the less abundant, with high specificity and sensitivity. Starting points for RNA-seq bioinformatics analysis include alignment-based methods, such as Bowtie [59], and STAR [71], or alignment-free methods, such as kallisto [72] and Salmon [73]. Cancer-related omics experiments often rely on specific, tailor-made analytic pipeline. TCGA and other repositories give the great opportunity to analyze the omics data by a pan-cancer approach where different types of cancers can be compared in terms of genomic and transcriptomic landscapes [74]. 

#### 3.1.4. Proteomics and Metabolomics

Given the high complexity and dynamic range of proteins, their identification and quantification in large scale are significantly challenging. Proteomic analyses are applied to identify and quantify the set of proteins present within a biological system of interest. Progressions of the tandem mass-spectrometry (LC-MS/MS) techniques in terms of resolution, accuracy, quantitation, and data analysis have made it a solid instrument for both the identification and quantification of cells proteome [75]. Recently, the advent of cutting edge high-resolution “Orbitrap” mass-spectrometer instruments associated with powerful computational tools (i.e., MaxQuant [76] and Perseus [77]) simplified the genome-wide detection of all expressed proteins in human cells and tissues paving the way for a first draft of the human proteome [78,79]. MS-based proteomics techniques have been extensively applied also to investigate the proteome alteration in several human cancer tissues [80]. In particular, the study of cancer proteomes is a promising path for biomarkers and therapeutic targets identification because proteins are the molecular unit from which cellular structure and function arise [81].

The application of MS techniques is not restricted to proteomics but rather can be extended to smaller molecules such as metabolites. Metabolomics is characterized by the quantifications of metabolites that are synthesized as products of cellular metabolic activities, such as amino acids, fatty acids, carbohydrates, and lipids. Their levels can be dynamically altered in disease states reflecting aberrant metabolic functions in complex disorders like cancer. Indeed, metabolic variations are significant contributors to cancer development [82]. This is the reason why cancer metabolomics has become an important research topic in oncology [83], with the aim to get new insights on cancer progression and potential therapeutic targets. Lipidomics is a subset of metabolomics [84], specifically cancer lipidomics has recently led to the identification of novel biomarkers in cancer progression and diagnosis [85]. Metabolomics is still an ongoing field with the potential to be highly effective in the discovery of biomarkers, especially in cancer. This is possible due to the support of bioinformatics tools like metab package [86], which provides an analysis pipeline for metabolomics derived from gas chromatography-MS data, or metaRbolomics package [87], which is a general toolbox that goes from data processing to functional analysis. Similarly, the lipidr package [88] is an analogous framework focused on lipidomics data processing.

### 3.2. Data Management

The huge amount of data deriving from different omics analyses need to be adequately collected and stored. Challenges of data management include defining the type of data to be stored and how to store it, the policies for data access, sharing, use, and finally, long-term archiving procedures [89]. One of the most successful repositories regarding application of multi-omics approach in cancer is NIHs Genome Data Commons (GDC) [90] containing all data generated by the Cancer Genome Atlas (TCGA) project [74]. TCGA project has performed integrative analysis of more than 30 human cancer types with the aim to create a publicly available comprehensive platform for collecting the molecular alterations in the cancer cells at the forefront of multi-omics research [74]. Information about aberrations in the DNA and chromatin of the cancer-genomes from thousands of tumors have been catalogued by matching with the normal genomes and linking these aberrations to RNA and proteins levels. Moreover, it provides data for method development and validation usable in many current projects. In 2020, the collaboration of an international team has completed the most comprehensive study of whole cancer genomes, significantly improving the fundamental understanding of cancer, and indicating new directions for developing diagnostics and treatments. The ICGC/TCGA Pan-Cancer Analysis of Whole Genomes Project (PCAWG, or the Pan-Cancer Project) involved more than 1300 scientists and clinicians from 37 countries, analyzed more than 2600 whole genomes of 38 different tumor types. Commenting this aspect, Rameen Beroukhim, an associate member of the Broad Institute, said: “It was heartening that this very large group was able to bring together disparate resources and work to come up with some groundbreaking findings”. Additionally, Gad Getz, an institute member and the director of the Cancer Genome Computational Analysis Group at the Broad Institute, director of bioinformatics at the Massachusetts General Hospital’s (MGH) Cancer Center and professor of pathology at Harvard Medical School, said: “This large international effort shows the breadth of the types of research and new biological insight that are possible using whole cancer genome data”. He continued: “By analyzing the largest collection of whole cancer genomes studied thus far, we created the most comprehensive catalog of mutational signatures to date, this catalog can be used to understand the mechanisms that generate mutations and drive cancer in each patient” [91]. The Pan-Cancer Project improved and developed new methods for exploring not only exome, that represent the 1 percent of the genome, but, also, the remaining 99 percent of the genome, which includes regions that regulate the activity of genes.

With the genomics, epigenomics, and transcriptomics data from over 11,000 tumors representing 33 of the most prevalent forms of cancer, the Pan-Cancer Atlas represents an exceptional chance for a comprehensive and integrated analysis to extend our current knowledge of how a normal cell achieves cancer hallmarks. The pan-cancer analysis involving multi-omics data in combination with structured bioinformatics and statistical instruments provides an effective platform to recognize common molecular signatures for the stratification of patients affected by different cancer types and uncover shared molecular pathology of different cancer types for designing tailored therapies. Investigation of the massive amount of cancer-specific data deposited in TCGA requires special bioinformatics methods to mine biologically meaningful information. Several analytic and visualization platforms have been already developed to support the rapid analysis of TCGA data. For instance, cBioPortal provides the opportunity to visualize, analyze, and download large-scale cancer genomics data sets [92]. The impulse for open data in the field of biomedical genomics is important to make data available in public repositories for improving and accelerating scientific discovery, although there are ethical and technological challenges to be overcome. 

### 3.3. Data Integration

The need to integrate multi-omics data has led to the development of new theoretical algorithms and methods that are able to extract biologically significant information of clinical relevance.

Unsupervised data integration refers to the cluster of methods that draw an inference from of an unlabeled input dataset. Learning consists in detecting intrinsic regularities and relationships between the data, without any prior knowledge about the data itself. Examples of unsupervised techniques are matrix factorization methods, Bayesian methods, network-based methods, and multi-step analysis. CNAmet is a powerful multi-step integration tool for CNV, DNA methylation, and gene expression data [93]. The identification of genes which are synergistically regulated by methylation and CNV data, allow the understanding of biological process behind cancer progression.

Supervised methods involve the use of a dataset for which the phenotype label is known. In this way, when the system has learned a given task, it will be able to generalize, or to use the experience gained to solve problems that provide the same basic knowledge. Supervised data integration methods are built via information of available known labels from the training omics data. The most common supervised techniques are Network-based methods, Multiple Kernel Learning methods, and multi-step analysis. For example, Feature Selection Multiple Kernel Learning (FSMKL) is a method which uses the statistical score for feature selection per data type per pathway, improving the prediction accuracy for cancer detection.

Semi-supervised integration methods, lies between supervised and unsupervised methods, takes both labeled and unlabeled samples to develop learning algorithm. It is particularly useful in cases where we have a partial knowledge about the data, or if the collection and sampling phase of labeled data is too expensive to be carried out exhaustively. Semi-supervised data integration methods are usually graph-based. Graph-based semi-supervised learning (SSL) methods have been applied to cancer diagnosis and prognosis predictions.

The combination of different biological layers, with the aim to discover a coherent biological signature, remain a challenging process. Furthermore, multi-omics combinations are not necessarily capable to achieve better diagnostic results. Selecting an optimal omics combination is not trivial, since there are economic and technical constraints in the clinical setting in which such diagnostic tools are to be deployed [94]. Machine Learning Bioinformatic approaches play an important role in the design of such studies.

#### 3.3.1. Multi-Omics Datasets

Selecting an appropriate dataset that allows for easy manipulation and data calculations could affect the performance of a computational model and reduce the main obstacles to multi-omics data analysis by improving data science applications of multiple omics datasets:The MultiAssayExperiment Bioconductor database [95] contains the information of different multi-omics experiments, linking features, patients, and experiments;The STATegRa dataset [96] has the advantage of allowing the sharing of design principles, increasing their interoperability;MOSim tool [97] provides methods for the generation of synthetic multi-omics datasets.

#### 3.3.2. The Problem of Missing Data

Integrating large amounts of heterogeneous data is currently one of the major challenges in systems biology, due to the increase in available data information [98]. The problem of missing and mislabeled samples, is a common problem in large-scale multi-omics studies [99]. It is common for datasets to have missing data related to some individuals. This often happens in clinical studies, where patients can forget to fill out a form. In other cases, it is possible that the acquisition of data reveals to be too expensive, need much time to be obtained or it is difficult to measure. Missing row values for a table are difficult to manage because most statistical methods cannot be applied directly to incomplete datasets. In recent years, several approaches have already been proposed to address missing row values [100]. The missRow package combines multiple imputation with multiple factor analysis to deal with missing data [99]. The omicsPrint method detects data linkage errors and family relations in large-scale multiple omics studies [101].

#### 3.3.3. Exploratory Data Analysis

Understanding the nature of the data is a critical step in omics analysis [102]. For this purpose, it is possible to use exploratory data analysis (EDA) techniques which allow better assessments at a further modeling step. The main techniques for EDA include cluster analysis and dimension reduction, both widely applied to transcriptomics data analysis [103]. While *cluster analysis* consists of a set of methods for grouping objects into homogeneous classes, based on measures related to the similarity between the elements, *dimension reduction* is the process of reducing the number of variables, obtaining a set of variables called “*principal*.” Both cluster analysis [104] and dimension reduction [105] are applied to cancer studies, as shown in Table 2.

Together with dimensionality reduction and data clustering, data visualization is also an important part of EDA [2]. The combinations of these three factors make it possible to identify complex patterns, subpopulations within a dataset, and understand the variability within a phenomenon. Even if the scatter plot is the most common method for data visualization, there are other visualization tools available. Hexbins [111] can be used to explore sc-RNAseq data, while Circos diagram [112] can be used for the detailed representation of multi-omic data and their position in specific genomic regions.

Recently it is stated that mapping omics data to pathway networks could provide an opportunity to biologically contextualize the data. A network representation of multi-omics data can enhance every aspect of the multi-omics analysis because the functional level of biological description is fundamentally composed of molecular interactions [2]. The main tools for a network representation of multi-omics data are Pathview [113] and Graphite [114].

#### 3.3.4. Machine Learning Models

In recent years, machine learning has been proved to be capable of solving many biomedical problems. These mathematical models can represent the relationships between observed variables and provide a useful description of biological phenomena. A ML tool can perform several tasks, including classification task in which the input data are divided into two or more classes and the learning system produces a model capable of assigning one class among those available to each input. These models have important biomedical applications [94], because they are capable of discriminating between health and disease, or between different diseases outcomes [2]. In a regression task instead, the output belongs to a continuous rather than discrete domain. These models provide insights into the molecular mechanisms driving physiological states, reveal interactions between different omics, and have been used in prognostic tools [115]. In this context, due to the large amounts of heterogeneous data, the removal of non-informative characteristics which simplifies the model, increases its performance, and makes it less expensive to measure, reveals to be a crucial process [2]. Feature selection algorithm is a process which selects the variables that contribute most to the prediction, removing the irrelevant or less important features that can negatively contribute to the performance of the model. Both classification and regression ML techniques combined with feature selection algorithms have been widely used for cancer prognosis and prediction [2]. Moreover, many packages, which combine exploratory, supervised, and unsupervised tools, have been recently implemented in oncology. Table 3 provides a list of some of these new tools.

#### 3.3.5. Functional Enrichment Approaches

The interpretation of a ML model results could be a difficult task. A strategy that can provide readily interpretable results consist in mapping omic data on functional characteristics, in order to make them more informative and to associate them with a wider body of biomedical knowledge [2]. Some functional enrichment approaches are listed below:Over-Representation Analysis (ORA) [124];Gene-Set Enrichment Analysis (GSEA) [125];Multi-Omics Gene-Set Analysis (MOGSA) [126];Massive Integrative Gene Set Analysis (MIGSA) [127];Exploratory Data Analysis (PCA) [128];Divergence Analysis [129].

The first two enrichment approaches, ORA and GSEA, are feature extraction methods generally employed as dimensionality reduction methods. The output of these methods could be the starting points for more complex models such as interactions among functions. In particular, ORA method is based on a statistical evaluation of the fraction of pathway components found among a user-selected list of biological components. This input list fulfils the specific criteria (i.e., log fold change, statistical significance, and cutting-off the majority of components from the input list such as all the genes of a microarray experiment). GoMiner [130] is one of the most popular examples of ORA method. It was developed for gene-expression analysis of microarray data. It takes as input a set of over-/under-expressed genes plus the complete set list of the microarray, then it calculates over-/under-representation for Gene Ontology categories by means of Fisher’s exact test. Similarly, GSEA was developed for gene expression analysis from microarray data. The input is a list of ranked genes in accordance with their differential gene expression between two phenotypic classes. For each set of genes, an enrichment score (ES) is calculated based on a Kolmogorov–Smirnov pathway-level statistic. Multiple hypothesis testing is applied for the evaluation of ES significance. In the study of [131], the GSEA methodology was used to validate the proliferative role of growth-supporting genes involved in cancer treatment [132]. Multi-omics gene-set analysis (MOGSA) is an enrichment approach that uses multivariate analysis, which consists in integrating multiple experimental and molecular data types measured on the same data set. The method projects the features across multiple omics data sets to reduce dimensional spaces and calculates a gene set score with the most significant features. MOGSA’s multi-omics approach compensates for missing information in each single data type to find sets of genes not obtainable from the analysis of single omics data. A different approach is the massive integrative gene set analysis (MIGSA). It allows to compare large collections of datasets from different sources and create independent functional associations for each omic layer. The utility of MIGSA was demonstrated in [133] by applying the multi-omics perspective method to functionally characterize the molecular subtypes of breast cancer. There are enrichment approaches, such as pathwayPCA and divergence analysis methods, which use functional aggregation as support for other data analysis studies. In pathwayPCA, exploratory data analysis is performed using statistical methodologies to analyze the functional enrichment of each omics set and aggregating them via consensus. pathwayPCA overcomes alternative methods for identifying disease-associated pathways in integrative analysis. Among various case studies, the model was applied for the identification of sex-specific pathway effects in kidney cancer for the construction of integrative models for the prediction of the patient’s prognosis and for the study of heterogeneity in an ovarian cancer dataset. Divergence analysis method instead, is an enrichment approach that uses functional aggregation to classify large amounts of omics data. The omic profile is reduced to a digital representation based on that of a set of samples taken from a baseline population. The state of a subprofile that is not within the basic distribution is interpreted as “divergent.” In [134] an application of the divergence analysis within the study of metabolic differences among the interpersonal heterogeneous cancer phenotypes has been described.

## 4. Novelty, Challenges, and Future Perspective

The computational approach plays a central role in improving our current cancer diagnostic capabilities [135]. The understanding of the cancer progression, the new therapeutic interventions, and the discovery of novel cancer biomarkers need to adopt and integrate different omics strategies at multiple levels. To achieve this aim, as suggested in the work of [136] there are five essential challenges in the omics integration workflow: (1) experimental challenges, (2) individual omics datasets, (3) integration issues, (4) data issues, and (5) biological knowledge. 

Experimental challenges: an accurate sample preparation in a multi-omics perspective becomes one of the major experimental challenges, with the aim to achieve a universal sample collection and preparation protocol for generating multiple omics datasets.Individual omics datasets: data preprocessing is also another significant challenge. This process can be performed on each omic dataset independently before merging significant results or after the production of a unique merged dataset. Moreover, the information included in each individual omic dataset requires very different standardization and scaling approaches, operating in different numerical and time scales.Integration issues: data integration issues increases the difficulty of accounting for false positives in merged datasets. Additional problems include the management of rigorous approaches based on statistical models with respect to less rigorous approaches that include a biological interpretation. In comparison to a single omics study, a multi-omics approach has the benefit to allow a deeper understanding of how the tumoral transformation is affecting the flow of information from different omics levels resulting in a bridge between cancerous genotype and the phenotype.Data issues: the storage of omics data is very important for reproducibility. To this end, new omic platforms are being developed to provide essential clinical data for insights into the prognosis and diagnosis of diseases.Biological knowledge: the interpretation of the outputs of computational models requires a deep knowledge of the biological system under study, in order to discriminate results that are not biologically relevant.

Despite these challenges the application of bioinformatics data integration and analysis, as well as the use of molecular modeling algorithms, allow to formulate many predictions of drug–target interactions to greatly facilitate guided drug development and guided drug resistance prevention [137]. Artificial intelligence (AI) approaches act on many aspects related to cancer therapy, including drug discovery and development and how these drugs are clinically validated and ultimately administered to patients [138]. The convergence of AI and cancer therapy has led to multiple benefits in terms of cost and time reduction. AI methods, ranging from regression models to neural networks can accelerate drug discovery, harness biomarkers to accurately match patients to clinical trials, and truly customize cancer therapy using only patients’ own data. 

In conclusion, the design and development of methods that integrate different multi-omic computational approaches in order to create robust and reliable models can lead to enormous advances in understanding the biology of cancer. As bioinformatics tools evolve, they must become user-friendly, interconnected, interoperable, and powerful for intensive analyses. In this context, integrated omics is not just an ensemble of computational tools, but a cohesive paradigm for deeper biological interpretation of multi-omics datasets that will potentially reveal novel details into cancer investigation. Although this field is still under development, many advances are constantly being made, with the development of new updated algorithmic approaches.

## Figures and Tables

**Figure 1 ijms-22-05751-f001:**
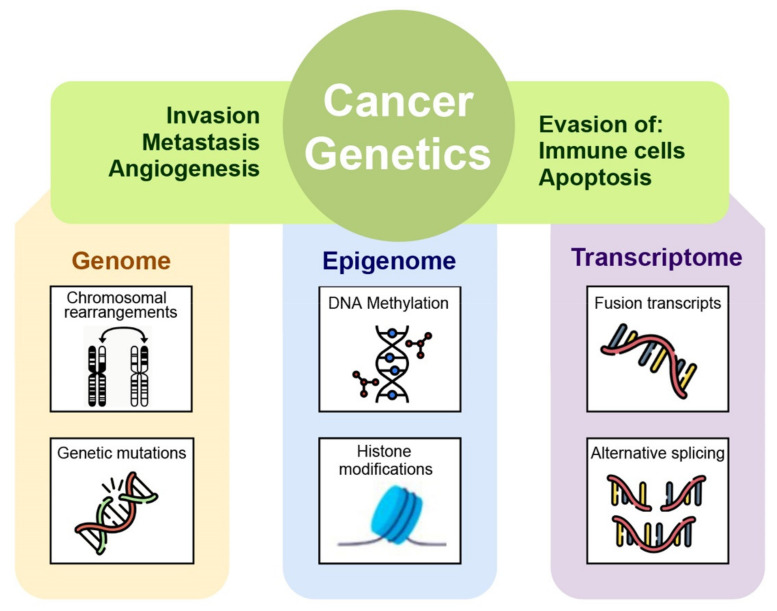
The many levels of interactions found in a cancer system, that can be measured via the different omics technologies, such as genomics, epigenomics, transcriptomic, and proteomic.

**Figure 2 ijms-22-05751-f002:**
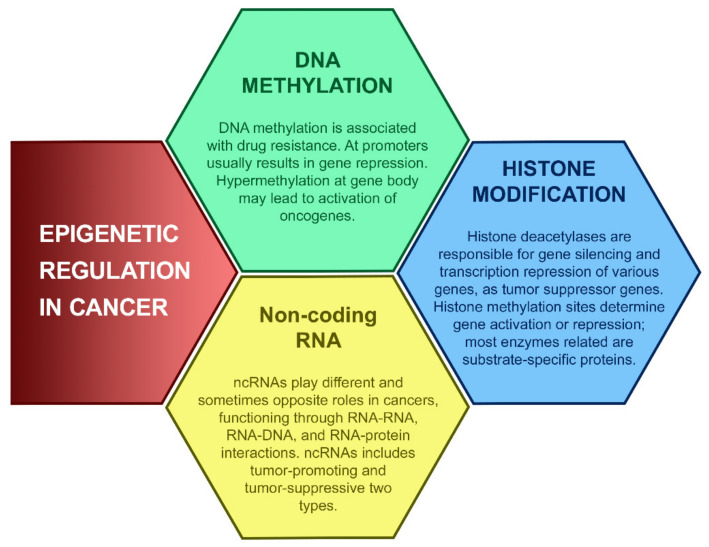
Epigenetic regulations in cancer. Alterations in epigenetic modifications in cancer regulate various cellular responses, including cell proliferation, apoptosis, invasion, and senescence. Through DNA methylation, histone modification, and noncoding RNA regulation, epigenetics play an important role in tumorigenesis. These main aspects of epigenetics present reversible effects on gene silencing and activation via epigenetic enzymes and related proteins.

**Table 1 ijms-22-05751-t001:** Summary of the applications of different techniques for sequencing, which are at the basis of different omics data acquisition approaches. Genomics, epigenomics, and transcriptomics are based on NGS techniques, whereas proteomics and metabolomics are driven by mass-spectrometric (LC-MS/MS) method. The main goal of genomics, epigenomics, and transcriptomics is the screening of genome-wide mutations, the identification of altered epigenomic modifications, and exploring differential RNA expression, while for proteomics and metabolomics is the identification of differentially regulated proteins and metabolites (reprinted from reference [6]).

OMICS	TYPE	PRINCIPLE		APPLICATION	BIOINFORMATICS TOOLS
GENOMICS	Whole exome sequencing	NGS		Exome-wide mutational/analysis	BWA Bowtie Bowtie2 SNAP SAM BAM
	Whole genome sequencing	NGS		Genome-wide mutational/analysis	
	Targeted gene/exome sequencing	Sanger sequencing		Mutational analysis in targeted gene/exon	
EPIGENOMICS	Methylomics	Whole genome bisulfite sequencing		Genome-wide mapping of DNA methylation pattern	Methylation-Array-Analysis SICER2 PeakRanger GEM MUSIC PePr DFilter MACS
	ChIP-sequencing	NGS		Genome-wide mapping of epigenetic marks	
TRANSCRIPTOMICS	RNA-sequencing	NGS		Genome-wide differential gene expression analysis	Bowtie STAR kallisto Salmon
	Microarray	Hybridization		Differential gene expression analysis	
PROTEOMICS	Deep-proteomics	Mass-spectrometry	Differential protein expression analysis		MaxQuant Perseus
METABOLOMICS	Deep-metabolomics	Mass-spectrometry	Differential metabolite expression analysis		Metab metaRbolomics Lipidr

**Table 2 ijms-22-05751-t002:** Main cluster analysis and dimension reduction package tools applied to cancer studies.

Package Tools	Description
OMICsPCA	Omics-oriented tools for PCA analysis [106]
CancerSubtypes	Contains clustering methods for the identification of cancer subpopulations from multi-omics data [107]
Omicade4	Implementation of multiple co-inertia analysis (MCIA) [108]
Biocancer	Interactive multi-omics data exploratory instrument [109]
iClusterPlus	Integrative cluster analysis combining different types of genomic data [110]

**Table 3 ijms-22-05751-t003:** Main packages tools implemented in oncology for machine learning.

Package Tools	Description
mixOmics	R package for the multivariate analysis of biological datasets with a specific focus on data exploration, dimension reduction, and visualization [116].
DIABLO	Package for the identification of multi-omic biomarker panels capable of discriminating between multiple phenotypic groups. It can be used to understand the molecular mechanisms that guide a disease [117].
MOFA	Package for discovering the principal sources of variation in multi-omics data sets [118].
Biosigner	Package for the identification of molecular signatures from large omics datasets in the process of developing new diagnostics [119].
omicRexposome	Package that uses high-dimensional exposome data in disease association studies, including its integration with a variety of high-performance data types [120].
OmicsLonDA	Package that identifies the time intervals in which omics functions are significantly different between groups [121].
Micrographite	Package that provides a method to integrate micro-RNA and mRNA data through their association to canonical pathways [122].
pwOmics	Package for integrating multi-omics data, adapted for the study of time series analyses [123].

## Data Availability

Not applicable.

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
