# Peer review of "Multi-Omics Model Applied to Cancer Genetics"

_ijms, 2021, doi:10.3390/ijms22115751_

Round 1

Reviewer 1 Report

The manuscript submitted by Pettini et al discusses one of the most important topic of today. There has been an increasing interest and potential for the multi-omics analyses of cancer datasets. Thus, shedding light on the multi-omics model is certainly of great advancement. However, being an evolving field, it needs a careful consideration. Authors are suggested to address following comments:

  • This manuscript is a good fit as a book chapter than the review article. Many of the text looks like an introduction chapter to the PhD thesis. Importantly, a review article discusses a number of studies in the field and provides a conclusive statement, rather than giving an extensive theoretical background.
  • One of the biggest concern is the figures which are used in this manuscript. Do authors have permission for the respective sources to reproduce these figures? Authors are strongly suggested to avoid existing figures from previously published literature. One of the characteristics of review articles is that it defines novelty and provides perspective. Such redundancy in the manuscript will affect both of these aspects.

Author Response

We thanks the reviewer for his/her constructive comments, since we decided to request to the editor permission to edit a little changing in the title (“ Multi -Omics Model Applied To Cancer Genetics”) and the abstract, to make it better suitable for the topic.

Point 1 - We have included more studies in the field and we have also greatly shortened the beginning theoretical background. The introduction section was shortened as suggested by the reviewer. However, because of the high heterogenicity of backgrounds in people approaching precision medicine, we believe that providing general theoretical information on genomics, epigenomics and transcriptomics to understand the phases of multi -omics approach in the genetic cancer research is essential.

Point 2 - Thank for your suggestion regarding the figures; they are covered by CC-BY-4 license, in open access papers, and we have followed the appropriate attribution credit. We have deleted Figure 2, since it was unessential. Figure 1 and Figure 3 are essential to give the background for understanding review topic. Figure 1 was edited with a more specific focus in a new “dress”. We would prefer not to change Figure 3 (source https://www.nature.com/articles/s41392-019-0095-0/figures/2), because, while we respect your point of view, we think that reuse of existent appropriate figures could encourage citations and acknowledgment of hard work of other research team, also considering the type of article and the section in which they are inserted.

Point 3 - In order to better fit with review article architecture, we have, also, provided a “Novelty, Challenges and Future Perspective” section.

Reviewer 2 Report

The authors attempt to present an overview of multi-omic data integration in cancer diagnosis, according to a translational point of view about perspective diagnostic tools. Several figures and tables support the main text, and authors should be praised for their work. However, whereas I agree with the statement that the "computational approach plays a central role in improving our current cancer diagnostic capabilities", I've found that in some parts the manuscript fails to correspond to the authors' aim.

The manuscript looks like constituted by two different portions, one represented by a (sometimes too deep) analysis of genetic and epigenomic alterations in cancers, and one that more clearly expresses the pivotal point of the article.

Despite the initial statement that "before starting to a deep insight in the bioinformatics approach and utilities involved in oncology, we must spend some words about the genetic connection", I think that section 2 "Genomics and Molecular Processes" is really too wide, and it must be greatly shortened, or appear as a separated manuscript. Different paragraphs are really common and well known among oncologists, or among people devoted to oncological basic researches, and should be substituted with some references.

On the contrary, both the aim of "consider what is a system biology framework and how to create a multi -omics model", or "the new frontiers in oncology-related to drug discovery and other aspects, like using RNA interference for diagnostic procedure", could be deeper investigated. 

Just a couple of examples:

-Table 1. Examples of main chromosomal translocations associated with different cancers. Is this table absolutely necessary within the main text? Whereas useful, it is not fully comprehensive, and it could divert readers' attention. I suggest moving it as Appendix if the Authors think it is essential to keep it.

-4.3.5. Functional Enrichment Approaches 

This subsection, which looks really interesting for the readers must be improved, maybe with some suggestions about indications and fields of application of the different models presented.

Author Response

Response

Point 1 - We thanks the reviewer for his/her constructive comments, since we decided to request to the editor permission to edit a little changing in the title (“ Multi -Omics Model Applied To Cancer Genetics”) and the abstract, to make it better suitable for the topic.

Point 2 - We have remodelled the architecture of the article, including more studies in the field, and we have greatly shortened the beginning theoretical background. The introduction section was cutted and adapted with paragraph 3. However, because of the high heterogenicity of backgrounds in people approaching precision medicine, we believe that providing general theoretical information on genomics, epigenomics and transcriptomics to understand the phases of multi -omics approach in the genetic cancer research is essential. We have deleted Figure 2, since it was unessential. Figure 1 and Figure 3 are essential to give the background for understanding review topic. Figure 1 was edited with a more specific focus in a new “dress”. We would prefer not to change Figure 3 (source https://www.nature.com/articles/s41392-019-0095-0/figures/2), because we think that reuse of existent appropriate figures could encourage citations and acknowledgment of hard work of other research team, also considering the type of article and the section in which they are inserted.

Point 3 – We, really, appreciate your example and we decided to move Table 1 as appendix and improved the subsection “Functional Enrichment Approaches”. Moreover, we have provided a “Novelty, Challenges and Future Perspective” section.

Round 2

Reviewer 1 Report

Authors are strongly suggested to include new figures designed by themselves. I have a strong concern about copyrights. Alternatively, authors shall attach "no-objection" letter from original authors of the respective figure(s) and their journal(s).

Also, why authors wants to remove subsections proteomics and metabolomics? Multi-omics do include both of these important levels of -omics. 

Author Response

Point 1 – Thanks for the suggestion. Now, all figures are designed by us.

Point 2 - We agree with the reviewer for this comment about the importance of these two levels of –omics. As suggested, we have reintroduced the proteomics and metabolomics subsections.

Reviewer 2 Report

The authors had modified the previous version, and I think that the manuscript is now suitable for publication in the present form

Author Response

Point 1 – Many thanks for your support; we really appreciate your constructive comments in this review report.
